# Single-cell transcriptomes reveal a molecular link between diabetic kidney and retinal lesions

Ying Xu[1], Zhidan Xiang[1], Weigao E[2], Yue Lang[1], Sijia Huang[3], Weisong Qin[1], Jingping Yang [1,4✉], Zhaohong Chen [1✉] & Zhihong Liu [1✉]

The occurrence of diabetic nephropathy (DN) and diabetic retinopathy (DR) are closely associated in patients with diabetes. However, the cellular and molecular linkage of DN and DR has not been elucidated, and further revelations are needed to improve mutual prognostic decisions and management. Here, we generate and integrate single-cell RNA sequencing profiles of kidney and retina to explore the cellular and molecular association of kidney and retina in both physiological and pathological conditions. We find renal mesangial cells and retinal pericytes share molecular features and undergo similar molecular transitions under diabetes. Furthermore, we uncover that chemokine regulation shared by the two cell types is critical for the co-occurrence of DN and DR, and the chemokine score can be used for the prognosis of DN complicated with DR. These findings shed light on the mechanism of the co-occurrence of DN and DR and could improve the prevention and treatments of diabetic microvascular complications.

[1] National Clinical Research Center of Kidney Diseases, Affiliated Jinling Hospital, Medical School of Nanjing University, Nanjing, China. [2] Center for Stem Cell and Regenerative Medicine, The First Affiliated Hospital, Zhejiang University School of Medicine, Hangzhou, China. [3] National Clinical Research Center of Kidney Diseases, Jinling Hospital, School of Medicine, Southeast University, Nanjing, China. [4] Medical School of Nanjing University, Nanjing, China. ✉email: jpyang@nju.edu.cn; czh4@sina.com; liuzhihong@nju.edu.cn

Diabetic nephropathy (DN) and diabetic retinopathy (DR) are the most common and severe complications of diabetes. DN occurs in 20–40% of diabetic patients and is the leading cause of end-stage renal disease[1]. DR is the leading cause of blindness among the working-age population, and almost all individuals with type 1 diabetes and 60% of individuals with type 2 diabetes will develop DR 20 years after the onset of diabetes[2]. Epidemiological studies suggest that DN and DR are closely linked in patients with diabetes[3,4]. The probability of DN in patients with DR is three times higher than that in patients without DR[5], and the adjusted hazard rate of DR is more than five in patients with DN compared with patients without DN[6]. Additionally, DN and DR share common risk factors, including obesity[7,8], pathological changes such as thickening of the basement membrane[9,10], and local inflammation[11,12]. Although the presence of either DN or DR is suggested to be helpful for diagnosing the other[13,14], the accurate diagnosis of DN still requires renal biopsies[15,16]. The cellular and molecular basis for the observed association between DN and DR is not yet fully characterized, and further heterogeneity may exist, which may explain the difficulty in making accurate predictions. Understanding the link between the kidney and retina in patients with diabetes would improve the screening and treatment strategies for both diseases.

The kidney and retina are both important organs with multiple functions and complex structures. Single-cell RNA sequencing (scRNA-seq) enables the distinction of cell types within complex organs and the identification of cell type-specific molecular features[17]; thus, scRNA-seq provides an opportunity to reveal the underlying association between diseases. The integrated study of kidney and retina single-cell profiles under physiological and pathological conditions would resolve the underlying association between DN and DR and facilitate early diagnosis and treatment decisions.

In this study, we integrated scRNA-seq profiles of human kidneys and retinas to explore the cellular linkage between the two organs. We found a similar gene expression pattern between renal mesangial cells (MCs) and retinal pericytes (RPCs). Further validation of this cellular association under physiological and pathological conditions in mice confirmed this cellular linkage and revealed that MCs and RPCs shared the molecular feature of upregulated chemokine production under diabetic conditions. By profiling of the transcriptomes of a cohort of DN patients, we demonstrated that the chemokine score could be used to assess the prognosis of DN complicated with DR. Our results provide insights into the association between the kidney and retina and the cellular and molecular basis of the cooccurrence of DN and DR.

## Results

### ScRNA-seq profiles of human kidney and retina reveal similarity between MCs and RPCs.
To explore the cellular linkage between the kidney and retina, we integrated scRNA-seq profiles of the human healthy kidney[18] and retina[19]. After quality control, a total of 21,238 cells from kidneys and 9264 cells from retinas were collected. After batch effect removal and dimension reduction, we successfully merged the cells from the two tissues. In total, we identified 47 clusters that included all the main cell types in the kidney and retina, such as glomerular endothelial cells (EC-GCs), MCs, podocytes (PODs), retinal endothelial cells (EC-Rs) and RPCs (Fig. 1a). The cellular landscape provides a comprehensive map to study the cellular relationship between the kidney and retina. Although ECs were present in both the kidney and retina, we found that EC-GCs and EC-Rs were separated and clustered into different subsets, indicating

tissue-specific heterogeneity of ECs. In contrast, we found that MCs clustered together with RPCs (Fig. 1b, Supplementary Figs. 1 and 2). We further examined marker genes of MCs or RPCs. The results showed that RPCs expressed most MC markers and vice versa (Fig. 1c). For example, expression of RGS5, ACTA2 and MYH11 were shared mainly in MCs and RPCs but rarely in other cell types (Fig. 1d). Cultured MCs were verified to express the RPC marker PDGFRB and cultured RPCs were verified to express the MC marker ACTA2, while RGS5 was shown to be a common marker (Supplementary Fig. 3). These results demonstrate that ECs from different tissues were heterogeneous, but MCs and RPCs showed a high degree of transcriptional similarity, indicating a potential cellular linkage between the kidney and retina.

### MCs and RPCs share molecular features under both physiological and diabetic conditions.
To confirm the similarity between MCs and RPCs, we further performed scRNA-seq of MCs and RPCs exposed to both physiological and pathological conditions. The db/db mouse model is one of the best models for studying type 2 diabetes, and it is used to study both DN and DR[20,21]. C57BLKS/J mice serve as the control as they have the same genetic background as db/db mice. Thus, we first generated scRNA-seq profiles from C57BLKS/J mice ($n = 3$ for kidney and $n = 8$ for retina) to investigate cells under physiological conditions.

We isolated glomeruli and retinas, followed by cell sorting of retinal vascular cells to enrich RPCs. With 10× Genomics Chromium, we captured 7277 cells from glomeruli and 6369 cells from retinal vessels. The glomerular cells were grouped into 7 clusters: EC-GCs, MCs, PODs, B cells, T cells, neutrophils and macrophages. Retinal vascular cells were grouped into 2 clusters: EC-Rs and RPCs. We then merged the cells and analyzed them together (Fig. 2a). The results showed that MCs and RPCs were grouped together more than any of the other cell types analyzed, indicating that MCs and RPCs are more similar to each other than to other vascular cell types (Fig. 2b). We examined well-established cell type-specific markers from the cell marker website (http://biocc.hrbmu.edu.cn/CellMarker/). The results showed that MCs and RPCs shared the expression of markers, which distinguished them from the other cell types (Fig. 2c). For example, chondroitin sulfate proteoglycan 4 (CSPG4), which was seldom mentioned in MCs, was exclusively expressed in MCs and RPCs (Fig. 2d, Supplementary Fig. 4). Furthermore, we found that many of the marker genes shared by MCs and RPCs are related to extracellular matrix (ECM) composition, such as Cd34, Cspg4 and Mcam. The ECM is important for the stability of MCs, RPCs and PODs[22,23]. When we examined the genes that encode collagen, an important component of the ECM, we found that MCs and RPCs mainly expressed Col4a1 and Col4a2, while PODs expressed Col4a3 and Col4a4 (Fig. 2e). All these results confirmed that MCs and RPCs shared molecular features under physiological conditions.

To further investigate the cellular and molecular linkage of the kidney and retina under diabetic conditions, we then generated scRNA-seq profiles of glomeruli and retina from db/db mice ($n = 3$ for kidney and $n = 8$ for retina). We collected cells at 21 weeks when the mice had developed DN and DR (Supplementary Fig. 5). We captured 7709 cells from glomeruli and 8048 cells from retinal vessels. The MCs and RPCs still clustered together, suggesting potential and consistent molecular transitions in the two cell types under diabetic conditions (Fig. 3a, b). When we performed Gene Set Enrichment Analysis (GESA) to study the changes in these cells under diabetic conditions, we found that both MCs and RPCs upregulated the expression of genes related

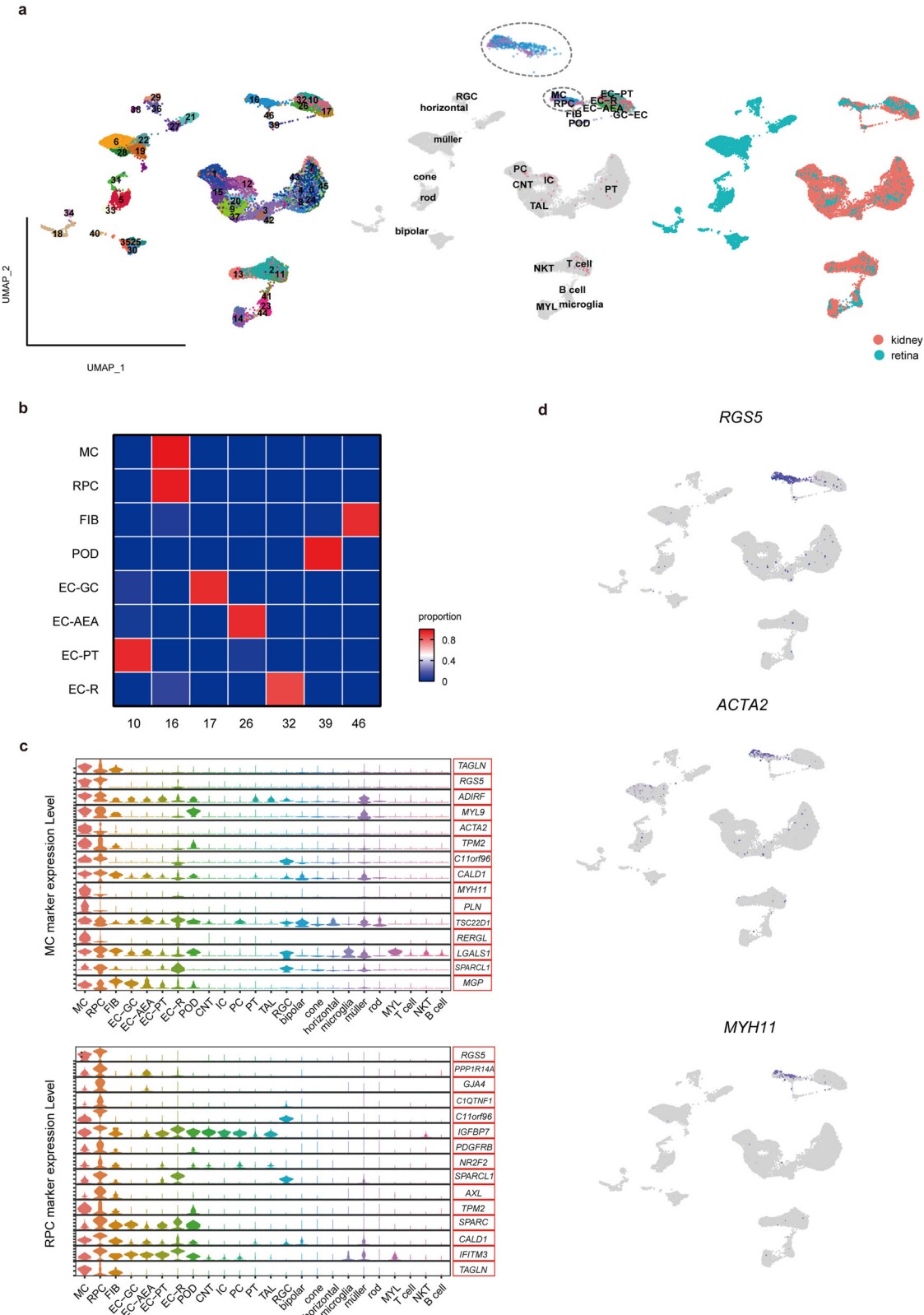

to the signaling pathways of ECM-receptor interaction and PI3K-Akt signaling pathway, while some energy metabolism pathways such as TCA cycle were upregulated only in RPCs (Fig. 3c, Supplementary Data 6). Interestingly, both MCs and RPCs underwent obvious remodeling of collagen composition. Under physiological conditions, MCs and RPCs mainly expressed *Col4a1* and *Col4a2*. However, under diabetic conditions, both cell types shifted to express other collagen components (Fig. 3d). MCs under diabetic conditions shifted to express *Col11a1* and *Col14a1*, which contributed to kidney fibrosis in DN[24,25]. RPCs increased the expression of *Col8a1*, *Col14a1*, *Col16a1* and *Col18a1*, among them, *Col18a1* has been reported to be associated with macular edema, neovascularization, and retinal detachment in DR[26]. These results suggest that MCs and RPCs not only share

**Fig. 1 ScRNA-seq reveals the cellular association between the kidney and retina. a** UMAP plot of 22,262 renal cells and 9711 retinal cells color-coded according to the number of clusters. Each dot represents a single cell and coloring is according to the unsupervised clustering performed by Seurat. Labeled numbers represent different cell clusters visualized by UMAP. The clusters annotated as MC, RPC, EC, POD and FIB are highlighted and the sample types of origin are shown. **b** Annotations of specific clusters are displayed. The heatmap displays the distribution of cell types within each cluster. MC and RPC are grouped together in Cluster 16. **c** Violin plots showing the expression of marker genes of MC (top) and RPC (bottom). **d** Expression of *RGS5, ACTA2* and *MYH11* showing shared transcriptional patterns of MCs and RPCs. Blue indicates maximum gene expression, while gray indicates low or no expression of a particular gene in log-normalized UMI counts. MC, mesangial cell, RPC, retinal pericyte, FIB, fibroblast, EC-GC, glomerular endothelial cell, EC-AEA, arteriolar endothelial cell, EC-PT, endothelial cell in peritubular, EC-R, retinal endothelial cell, POD, podocyte, CNT, connecting tubule, IC, intercalated cell, PC, principal cell, PT, proximal tubule cell, TAL, thick ascending limb cell, RGC, retinal ganglion cell, bipolar, retinal bipolar cell, cone, cone cell, horizontal, retinal horizontal cell, rod, rod cell, MYL, myeloid cell, NKT, natural killer T cell.

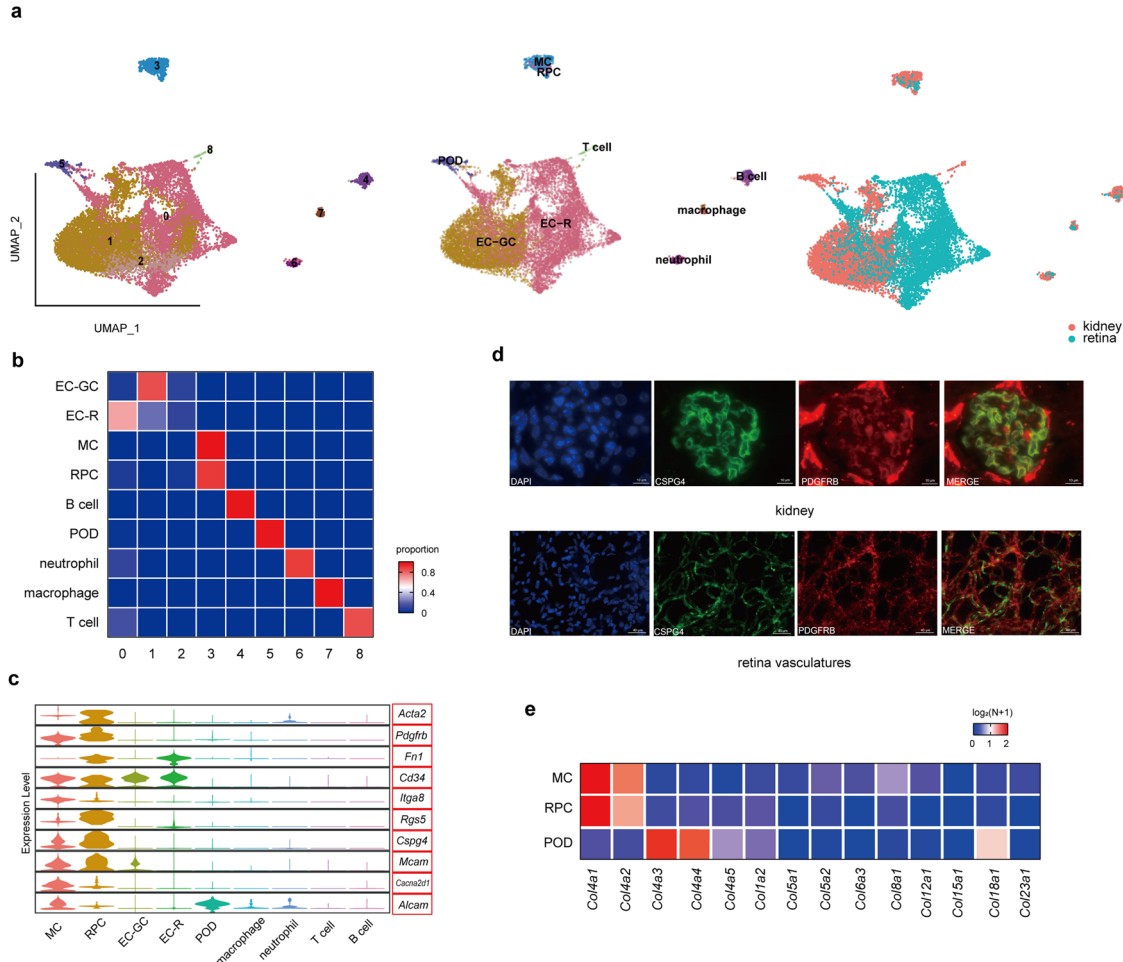

**Fig. 2 ScRNA-seq of mouse glomeruli and retinal vessels confirmed the similarity between MCs and RPCs. a** UMAP plot of 7277 cells from glomeruli and 6369 cells from retinal vessels. Each dot represents a single cell and the cell types and sample types of origin are annotated on the right. Each dot represents a single cell and coloring is according to the unsupervised clustering performed by Seurat. **b** Cluster annotations indicate the association between MCs and RPCs. The heatmap displays the distribution of cell types within each cluster. MCs and RPCs are grouped together in Cluster 3. **c** Expression of canonical marker genes of MCs and RPCs. The top 5 are markers of MCs, and the bottom 5 are markers of RPCs. **d** Immunofluorescence staining of CSPG4 (green), PDGFRB (red) and DAPI (blue). Colocalization of CSPG4 and PDGFRB confirmed the specific expression of CSPG4 in MCs and RPCs. Scale bar: 10 µm for glomeruli and 40 µm for retinal vessels. **e** Collagen composition of MCs, RPCs and PODs. Red indicates maximum gene expression, while blue indicates low or no expression.

transcriptional similarity under physiological conditions but also share molecular transitions in the context of DN and DR.

**Chemokine expression is upregulated in both MCs and RPCs.** To eliminate the effects of other cell types or other factors in vivo, we validated the changes in MCs or RPCs in vitro by treating these cells with advanced glycation end products (AGEs) to mimic diabetic conditions. We conducted RNA-seq on cultured human MCs (HMCs) and RPCs (HRPCs) treated with 100 µg/ml

AGEs or bovine serum albumin (BSA, as a control). The transcriptome of cells treated with BSA showed that HMCs and HRPCs shared 80% of the highly expressed genes (Supplementary Data 4), which were enriched in pathways related to cell adhesion and ECM organization (Fig. 4a, b), consistent with the cellular similarity revealed in vivo.

Both HMCs and HRPCs showed transcriptional changes after treatment with AGEs (Fig. 4c, Supplementary Data 5), in particular, changes were observed in genes related to

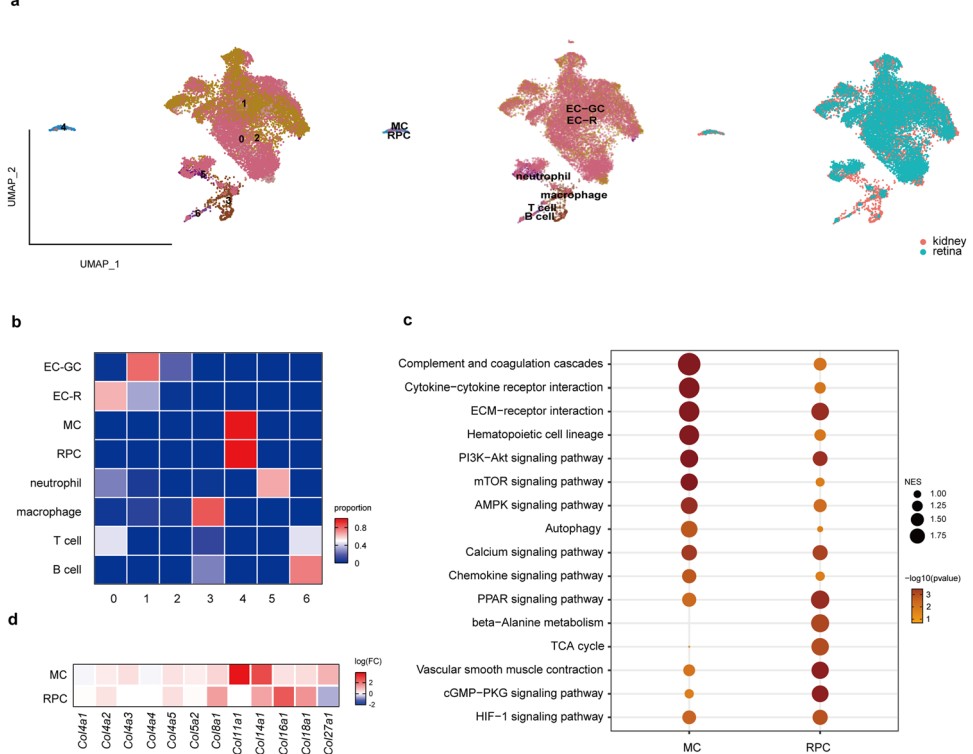

**Fig. 3 MCs and RPCs share molecular transitions under diabetic conditions. a** UMAP plot of 7709 cells of glomeruli and 8048 cells of retinal vessels from 21-week-old *db/db* mice. Each dot represents a single cell and the cell types and sample types of origin are annotated on the right. **b** Annotation of clusters shows the linkage of MCs and RPCs in *db/db* mice. The heatmap displays the distribution of cell types within each cluster. MCs and RPCs are grouped together in Cluster 4. **c** GESA results show the upregulated pathways in MCs and RPCs under diabetic conditions. The size of circle represents NES value. NES, normalized enrichment score. The darker color of circle indicates a smaller *p*-value. **d** Differential expression of genes encode collagen in MCs and RPCs under control and diabetic conditions. Red indicates a high fold change, while blue indicates a low fold change.

chemotaxis, such as *CXCL1*, *CXCL2*, *CXCL3*, *CXCL5*, *CXCL6* and *CXCL8*. We confirmed the upregulated expression of these chemokines at mRNA and protein levels by real-time PCR and ELISA (Fig. 4d, Supplementary Data 6). KEGG pathway enrichment of the differentially expressed genes (DEGs) confirmed that the TNF signaling pathway, cytokine-cytokine receptor interaction and chemokine signaling pathway were the top upregulated pathways shared by HMCs and HRPCs (Fig. 4e, Supplementary Data 6). Then, using the expression of chemokine genes from the molecular signature database, we defined chemokine scores of cells by the single-sample GSEA function built in the GSVA and found that the chemokine scores of MCs and RPCs increased in *db/db* mice (Fig. 4f, Supplementary Data 6), indicating the upregulation of chemokines in vivo. Previous studies suggest that these upregulated chemokines play roles in attracting neutrophils and macrophages[27] and aggravate tissue damage in patients with DN and DR[28]. When we examined the proportions of immune cells in glomeruli under physiological and pathological conditions, we found that the proportions of neutrophils and macrophages were increased (Fig. 4g). Furthermore, we co-stained MCs and RPCs marker CSPG4 with CXCL1 in kidneys of healthy and diabetic humans and mice (Fig. 5). We have identified an upregulation of chemokines in DN patients, and also noticed a partial co-localization of these chemokines with MCs. All these results suggested that diabetic conditions triggered the upregulation of genes related to chemokine signaling in both MCs and RPCs in DN and DR, and these phenomena might subsequently increase inflammation in kidney and retina.

**Chemokine score distinguishes DN patients with or without DR**. As we showed that chemokine production in MCs and RPCs could serve as a molecular and cellular linkage between DN and DR, it is possible that the chemokine score of MCs and RPCs could be used to improve the accuracy of the prognosis of patients with both DN and DR. To test this hypothesis, we utilized glomerular transcriptome data of 29 DN patients (14 patients complicated with unclassified DR and 15 patients without DR) and 20 healthy controls to investigate the association between chemokine scores and disease condition of patients, the clinical characteristics of DN patients were shown previously[29] and DN patients complicated with and without DR are detailed in Supplementary Table. 1. GSEA revealed that the expression of gene sets involved in the chemokine signaling pathway was increased in DN patients complicated with DR (adjusted $P = 3 \times 10^{-4}$) but not in those without DR (Fig. 6a). Furthermore, we calculated a chemokine score for each patient or control. We found that the chemokine scores of DN patients with DR were significantly higher ($P < 0.0001$) than those of DN patients without DR or healthy controls (Fig. 6b, Supplementary Data 6). To further examine the specificity of chemokine scores in distinguishing different clinical groups, we divided these patients and controls into 4 groups according to chemokine score quartiles: low, medium-low, medium-high and high. As the chemokine scores increased, the proportion of patients with DN complicated with DR also increased (Fig. 6c). The receiver operating characteristic (ROC) curve demonstrated the good performance of the chemokine score for distinguishing DN patients with or without DR. The area under the curve (AUC) of the chemokine score was 0.843 (Fig. 6d). To assess the

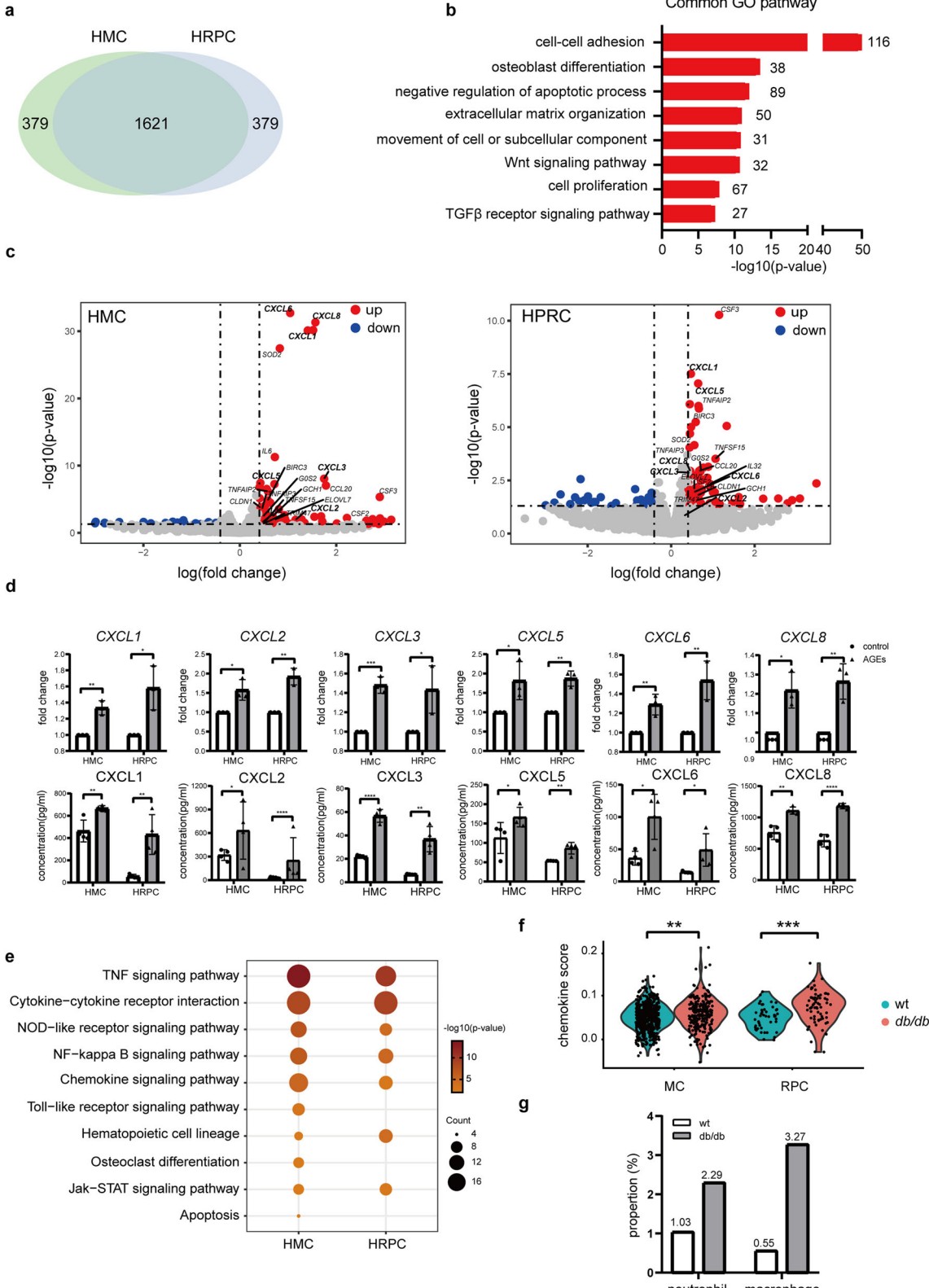

**Fig. 4 Transcriptomes of HMCs and HRPCs in response to control or AGEs treatment. a** The overlap of highly expressed genes in the normal control groups of HMCs and HRPCs. **b** Shared enriched GO pathways between HMCs and HRPCs, sorted by *p*-value. **c** Volcano plot shows DEGs in HMCs and HRPCs treated with AGEs. The red dots indicate upregulated genes while blue dots indicate downregulated genes. **d** Expression levels of *CXCL1*, *CXCL2*, *CXCL3*, *CXCL5*, *CXCL6* and *CXCL8* were measured by real-time PCR and ELISA between control and AGEs groups. **e** Significantly enriched KEGG pathways of DEGs in both HMCs and HRPCs treated with AGEs, the size of circle represents genes number enriched in pathways, the darker color of circle indicates a smaller *p*-value. **f** The chemokine scores of MCs and RPCs in wt and *db/db* mice. **g** The proportions of infiltrating neutrophils and macrophages in the glomeruli of wt and *db/db* mice. *P* < 0.05, **P* < 0.01, ***P* < 0.001, ****P* < 0.0001.

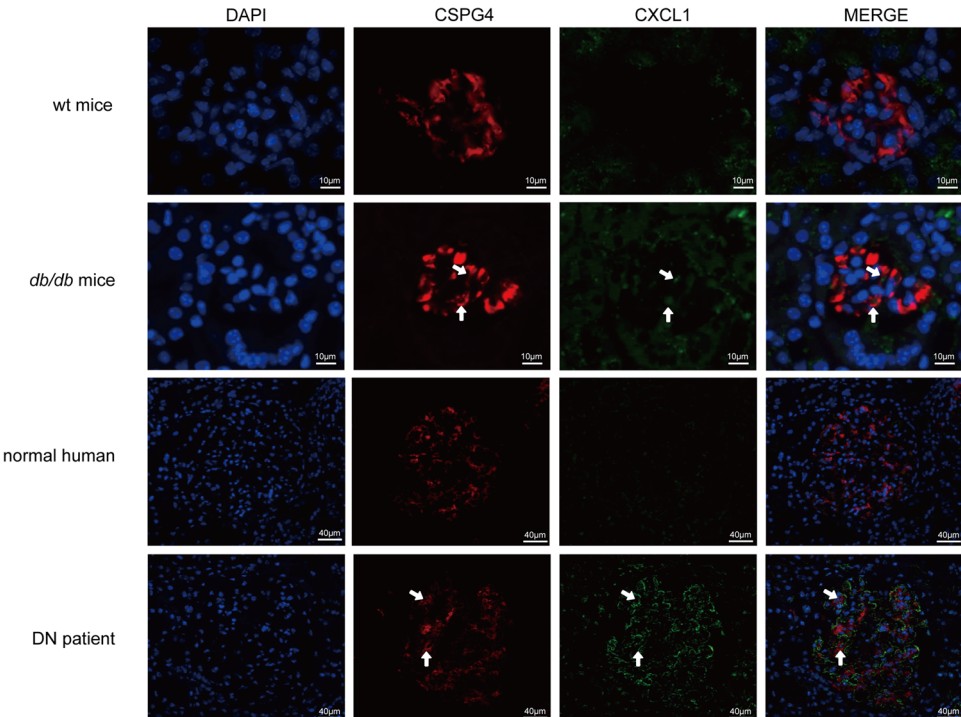

**Fig. 5 Verification of the increased expression of chemokines in DN by co-staining tissues of healthy and diabetic humans and mice with the MC and RPC markers CSPG4 and chemokine.** Co-staining of CSPG4 (red) and CXCL1 (green) in the kidney of wt and *db/db* mouse and in the kidney of normal human and DN patient. Scale bar: 10 μm for kidney in mice and 40 μm for kidney in humans.

progression of DN in patients of four groups, we performed a linear mixed-effects model and found that along with the increase in chemokine score, the estimating glomerular filtration rate (eGFR) declined faster (Supplementary Fig. 6). All these results demonstrate that the changes in chemokine expression shared by MCs and RPCs are critical for the linkage between DN and DR, and the chemokine score could be used for the prognosis of DN complicated with DR.

## Discussion

DN and DR are common microvascular complications of diabetes. DN is the primary cause of chronic kidney disease, accounting for the development of 40% of new cases of end-stage renal disease[30]. DR is the leading cause of blindness in working-age people[2]. As nephropathy and retinopathy are frequently linked in diabetic patients, understanding the mechanism underlying this linkage between DN and DR would improve early prediction and treatment decisions for DN complicated with DR.

The retina and kidney are both highly complex organs composed of specialized cell types. Our study explored the cellular relationship between the kidney and retina through scRNA-seq profiles. We found that MCs and RPCs shared similar molecular features. Consistent with our findings, MCs and RPCs are both reported to be mural cell types. They are located in blood vessels, are embedded within the basement membrane and are essential for proper microvascular function[31,32]. A previous study showed that RPCs exhibit functional similarity to MCs, as they can synthesize and release TGF-β, produce ECM and participate in the formation of the intraocular renin-angiotensin system[32]. In addition, it was observed that in diabetes, AGEs inhibit the proliferation of MCs and RPCs via the accumulation of a-series gangliosides, and this was hypothesized to be the common mechanism involved in the development of DN and DR[33]. In the present study, we revealed the transcriptional similarity between

MCs and RPCs in whole scRNA-seq profiles. The high degree of transcriptional similarity between MCs and RPCs may indicate that they are vulnerable to the same stimuli or undergo similar pathophysiological changes in the context of specific diseases.

We further identified the molecular signatures that are shared by MCs and RPCs in the context of diabetes. Enriched MCs and RPCs from *db/db* mice present similar pathological changes, such as upregulated expression of genes related to ECM-receptor interaction and PI3K-Akt signaling pathway, which can contribute to the progression of DN and DR[11]. The ECM provides both physical support to cells and regulated signaling[34,35]. In our study, we observed that MCs and RPCs express similar ECM components, indicating the presence of similar vascular microenvironments in the glomeruli and retinas. Moreover, we observed that the ECM is actively remodeled both in MCs and RPCs under diabetic conditions, and we identified the key ECM remodeling molecule Col11a1 in DN, which could probably serve as a new biomarker for altered ECM remodeling and provide exciting possibilities for therapeutic intervention.

In addition to ECM, we identified another molecular transition shared by MCs and RPCs, namely, the upregulation of chemokine production. Chemokines can attract neutrophils and macrophages and may be responsible for the increased numbers of inflammatory cells in glomeruli in diabetes. A growing body of evidence suggests that inflammatory cells play crucial roles in the pathogenesis of DN and DR. These inflammatory cells produce various proinflammatory cytokines and growth factors, which modulate the local response, increase inflammation and aggravate tissue damage[36,37]. Many studies have reported the importance of chemokines in DN or DR. The levels of chemokines positively correlate with the progression of DN[38]. Chemokines are also involved in the pathogenesis of diabetic macular edema as they affect the neurovascular unit[39]. Encouragingly, we found that the chemokine scores of DN patients with DR were significantly higher than those of DN patients without DR or healthy controls. A previous study found

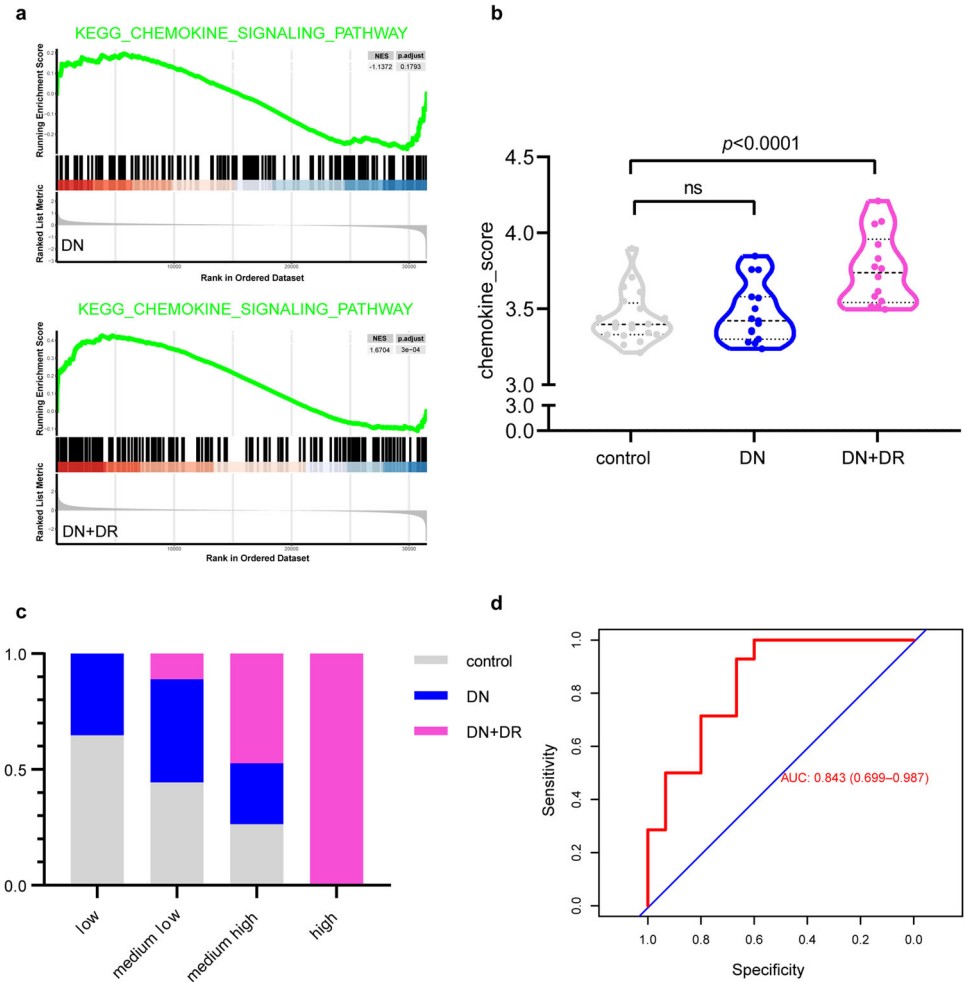

**Fig. 6 Chemokine scores of DN patients could distinguish patients with DN from those with DN and DR. a** GSEA of the expression of gene sets related to the chemokine signaling pathway in DN patients without DR (upper) or with DR (lower). **b** Chemokine scores were calculated according to the expression of chemokine genes. $n = 20$ for the control group, $n = 15$ for the DN group and $n = 14$ for the DN + DR group. **c** The proportion of control patients (gray), DN patients (blue) or DN complicated with DR patients (rose) in these groups. **d** AUC of the chemokine score of the logistic regression equation of the chemokine score.

that elevated chemokine levels could be considered biomarkers of DN[40]. Here, in our study, elevated chemokine scores also predicted the linkage between DN and DR, suggesting that chemokine levels could be a biomarker of the cooccurrence of DN and DR. The fundus examination collection of inpatients admitted to the nephrology department lacked exact grading of DR. This created difficulty in determining the correlation between chemokine scores and different DR levels, which should be improved in further research efforts. From the therapeutic perspective, several clinical studies have performed trials on inhibitors of chemokines or their receptors, such as CXCR2, in patients with DN or DR, and these studies have shown good preliminary results[41,42]. Thus, monitoring chemokine levels may help us to improve the early diagnosis of DN and DR. Additionally, chemokines can be potential therapeutic targets for DN and DR.

To summarize, our study has identified that the association between MCs and RPCs is a cellular basis for the cooccurrence of DN and DR. MCs and RPCs share molecular features under both physiological and diabetic conditions. Chemokines whose expression is elevated in MCs and RPCs under diabetic conditions could be biomarkers for the early detection of DR. This study provides a new perspective for exploring the relationship between the kidney and retina and lays the step for further studies on DN and DR pathogenesis, diagnosis and treatment.

## Methods

**Animals**. The animal experiments were approved by the Institutional Animal Care and Use Committee of Jinling Hospital (Nanjing, China), in accordance with the approved guidelines of the Institutional Animal Care and Use Committee of Jinling Hospital. 7 weeks old male wild-type (wt) and leptin receptor-deficient (*db/db*) mice on the C57BLKS/J background were purchased from the Model Animal Research Center of Nanjing University (Nanjing, China). All animals were maintained in rooms with constant temperature and humidity under a 12-h light/dark cycle. Body weight and fasting blood glucose levels were monitored biweekly with a glucometer. Murine urinary albumin and creatinine levels were measured using an Albuwell M (1011, Exocell, Philadelphia, USA) and Creatinine Assay Kit (dict-500, Bioassay systems, CA, USA) according to the manufacturer's instructions.

**Isolation of glomeruli and retinas and single-cell preparation**. Mice were sacrificed at 21 weeks of age on the basis of changes in the lesions of *db/db* mice[43,44]. Wt ($n = 3$) and *db/db* ($n = 3$) mice were anesthetized and first perfused with PBS, followed by Dynabeads (14013, Thermo Fisher Scientific, MA, USA) perfusion. Kidneys were harvested, minced, and digested with 1 mg/ml

collagenase II (C6885, Thermo Fisher Scientific, MA, USA), 1 mg/ml protease E (P6911, Thermo Fisher Scientific, MA, USA) plus 50 units/ml DNase I (D5025, Thermo Fisher Scientific, MA, USA) at 37 °C for 5 min with rotation (300 rpm). Separated glomeruli were washed three times, placed in 2 ml digestion buffer and incubated for an additional 30 min at 37 °C with rotation (1400 rpm). Glomerular cells were prepared after filtering the tissue through a 40 μm cell strainer and centrifuged at $300 \times g$ for 5 min at 4 °C. For the isolation of retinal cells, wt ($n = 8$) and $db/db$ ($n = 8$) mice were perfused with PBS, and globes were removed and dissected in PBS on ice. Retinal tissues were dissociated with a dissociation system containing 2 mg/ml collagenase II and 200 units/ml DNase I at 37 °C for 30 min with rotation (800 rpm) and triturated 10 times every 10 min. After being filtered through a 40 μm cell strainer and centrifuged at $300 \times g$ for 5 min at 4 °C, the retinal cells were sorted by magnetic bead separation using 5 μl PE-conjugated anti-mouse CD140b antibody (136006, BioLegend, CA, USA) and 5 μl PE-anti platelet and endothelial cell adhesion molecule 1 (PECAM1) antibody (160204, eBioscience, MA, USA). A total of 20 μl paramagnetic microbead-conjugated anti-PE antibody (130-048-801, Miltenyi Biotec, CA, USA) was used to separate the cells.

**ScRNA-seq**. Single-cell gel bead emulsions and libraries of glomerular and retinal vessel cell suspensions were prepared with a Chromium Single Cell 3' V3 Reagent Kit (PN-1000075, 10x Genomics, the Netherlands). Sequencing was performed on an Illumina (Illumina, USA) HiSeqX Ten System with a 150 bp paired-end read strategy.

**ScRNA-seq data processing and quality control**. Cell Ranger software (v3.0.0) from 10× Genomics was used to map reads to the genome and transcriptome to produce a matrix of gene counts versus cells. Seurat (v3.2.3) was used to process the unique molecular identifiers. To perform quality control, we retained cells with >200 genes and <5000 genes and discarded cells with <200 unique molecular identifiers or >30% mitochondrial RNA.

**scRNA-seq bioinformatics analysis**. The filtered expression matrix was normalized with the function "NormalizeData", followed by the identification of 2000 genes of high cell-to-cell variation by using the function "FindVariableFeatures". We then performed principal component analysis (PCA) with the top 2000 variable features by using the function "RunPCA". Cells were then clustered using the functions "FindNeighbors" and "FindClusters" with the first 50 principal components (PCs). Finally, UMAP was performed on the top 50 PCs by using the function "RunUMAP" for nonlinear dimensional reduction and data visualization. Then, we annotated the clusters based on canonical marker genes (Supplementary Table. 2, Supplementary Data 1–3) and integrated the kidney and retina data by Harmony.

**Differential gene expression and analysis of signaling pathways**. The DEGs were identified using DEsingle (v1.14.0). Genes with $P < 0.05$ and absolute log_2fold change >1.5 were used for functional enrichment analysis using the Gene Ontology (GO) database in clusterProfiler (v3.14.3). DAVID Bioinformatic Resources 6.8 was used to perform Kyoto Encyclopedia of Genes and Genomes (KEGG) analyzes (https://david.ncifcrf.gov/tools.jsp). In addition, ranked GSEA was performed with the Molecular Signature Databases MSigDB (v5.0). GSEA is supported by the Broad Institute website (http://www.broadinstitute.org/gsea/index.jsp).

**Gene set analysis**. Gene sets were obtained from the Molecular Signature Database MSigDB. The collagen gene set was integrated into C2: canonical pathway. The chemokine gene set was termed 'KEGG_CHEMOKINE_SIGNALING_PATHWAY'. We calculated the chemokine score of each clinical sample by evaluating the gene expression level of the chemokine gene set with the single-sample GSEA algorithm implemented in the Gene Set Variation Analysis (GSVA) package (v1.42.0).

**Human ScRNA-seq data sources and analysis**. The output files of the CellRanger (10× Genomics) pipeline of healthy human kidney (GSE140989) and expression matrix files of healthy human retina (GSE142449) were obtained from the NIH GEO dataset. Kidney samples were composed of 24 CryoStor preserved human kidney samples: 16 tumor-nephrectomy, 5 human allograft kidney transplant surveillance, and 3 preperfusion transplant biopsy samples. Kidney tissues were enzymatically digested at 37 °C after resuscitation. Human donor eyes ($n = 3$) were obtained through the Iowa Lionizers Eye Bank and were received in the laboratory within 5.5 h after death and were immediately subjected to enzyme dissociation at 37 °C and then cryopreserved. Both dissociated cells were processed with 10x Genomics and the Illumina HiSeq 4000 platform. The expression files were transformed to Seurat objects through the "CreateSeuratObject" function. Both datasets were normalized and scaled and PCA was performed using highly variable genes that were selected using the Seurat function "FindVariableFeatures". DoubletFinder (v2.0.3) was used to remove suspicious doublets. Cells were annotated with marker genes provided by original articles. We used Harmony to perform batch correction and data integration. The data were then visualized using UMAP.

**Human RNA-seq data sources and analysis**. Human glomeruli were obtained from the National Clinical Research Center of Kidney Diseases, Nanjing, China. The glomeruli were microdissected and subjected to RNA extraction, followed by cDNA synthesis and genome-wide gene expression analysis using the Affymetrix® microarray platform (HTA 2.0), as we previously described[29]. Chemokine score analysis included 29 biopsy-confirmed type 2 DN patients with fundus examination results and 20 healthy controls. Among all the DN patients, 14 individuals were identified as having DN complicated with DR, whereas the other 15 patients did not exhibit fundus changes. The exact DR status of these DR patients lacked due to the relatively simple description recorded in the inpatient medical record, so we grouped these patients into 2 groups: DN patients complicated with and without DR. The clinical features of these patients are listed in Supplementary Table. 1. DEGs were analyzed by using limma (v3.50.1). The predictive sensitivity and specificity of the chemokine scores in the clinical groups were assessed with ROC curves according to pROC (v1.18.0).

**Cell culture and treatment**. Human renal mesangial cells (HMCs) and human retinal pericytes (HRPCs) were purchased from Cell Systems company and maintained in DMEM (10567014, Thermo Fisher Scientific, MA, USA) supplemented with 10% FBS (10099141, Thermo Fisher Scientific, MA, USA). After they reached 70-80% confluency, the cells were cultured in medium supplemented with 0.5% FBS overnight for synchronization. The induction group was cultured in culture medium supplemented with 100 μg/ml AGEs (bs-1158p, Bioss, Beijing, China) for 6 h, and cultured medium supplemented with 100 μg/ml BSA (V900933, Sigma Aldrich, MO, USA) was used as a control.

**Bulk RNA sequencing**. Total RNA was extracted, and cDNA was generated with an RNA extraction kit (64-17-5, Takara, Japan)

according to the manufacturer's instructions. After amplification and purification, libraries were generated. Quality control was performed by a Qubit 3.0 Fluorometer (Thermo Fisher Scientific, MA, USA) and an Agilent 2100 bioanalyzer with a high-sensitivity DNA kit (Agilent Technologies, CA, USA). Sequencing was performed on a HiSeq3000 platform with a 150 bp paired-end read strategy to an average depth of 60 million reads per sample. DEG analysis was performed using DESeq2 (v1.20.0). Plots were generated in R with ggplot2 (v3.0.0) and pheatmap (v1.0.12).

**Real-time PCR**. Total RNA was extracted with an RNA extraction kit (64-17-5, Takara, Japan), and cDNA was synthesized with PrimeScript RT Master Mix (RR036A, Takara, Japan). Real-time PCR was performed with the 7900HT Sequence Detection System (Applied Biosystems, CA, USA), and the data were analysed using the $2^{-\Delta\Delta CT}$ method.

The primers are: Human *CXCL1*-forward: 5′-CAAACCGAA GTCATAGCCACAC-3′, reverse: 5′-ACCCTGCAGGAAGTGT CAATG-3′; Human *CXCL2*-forward: 5′-GAAAGCTTGTCTCAA CCCCG-3′, reverse: 5′-TGGTCAGTTGGATTTGCCATTTT-3′; Human *CXCL3*-forward: 5′-CCAAACCGAAGTCATAGCC AC-3′, reverse: 5′-TGCTCCCCTTGTTCAGTATCT-3′; Human *CXCL5*-forward: 5′-AGCTGCGTTGCGTTTGTTTAC-3′, reverse: 5′-TGGCGAACACTTGCAGATTAC-3′; Human 5′-*CXCL6*-forward: AGAGCTGCGTTGCACTTGTT-3′, reverse: 5′-GCAGTT TACCAATCGTTTTGGGG-3′; Human *CXCL8*-forward: 5′-ACT GAGAGTGATTGAGAGTGGAC-3′, reverse: 5′- AACCCTCTG CACCCAGTTTTC-3′.

**Immunofluorescence**. For immunofluorescence staining of cultured cells, HMC and HRPC cell lines were cultured in chambers. When the cells reached 80–90% confluence, the chambers were washed 3 times with cold PBS, and then, the cells were fixed with 4% paraformaldehyde for 15 min. After blocking with 10% fetal bovine serum (10099141, Thermo Fisher Scientific, MA, USA) for 30 min, the cells were incubated in a wet box overnight with the following primary antibodies: rabbit anti-PDGFRB antibody (1/ 100, ab32570, Abcam, UK), rabbit anti-αSMA antibody (1/100, ab32575, Abcam, UK), rabbit anti-RGS5 antibody (1/100, ab196799, Abcam, UK), rabbit anti-CXCL1 antibody (1/100, 12335-1-AP, Proteintech, China) and rabbit anti-CXCL3 antibody (1/100, YT2075, ImmunoWay, USA). All the cells were then incubated with FITC-conjugated rabbit anti-human IgG(H+L) (1/500, A0562, Beyotime Biotechnology, Shanghai, China) antibody and Cy3-conjugated goat anti-rat IgG(H+L) (A0521, Beyotime Biotechnology, Shanghai, China) for 1 h after washing with PBST 3 times. The chambers and kidney sections were stained with Antifade Mounting Medium with Hoechst 33342 (P0133, Beyotime Biotechnology, Shanghai, China) and imaged with a Zeiss confocal microscope.

Tissues from mouse kidneys were formalin-fixed, paraffin-embedded, and processed for sectioning. Retinal blood vessels ware prepared as previously described. Briefly, the retinas were separated under a microscope, placed in 1 mL of deionized water in a 24-well plate and shaken at 200 rpm with a 1.5 mm vibration orbit at room temperature until transparent. The sections were blocked with 10% bovine serum in PBS for 10 min and incubated with primary antibodies against CSPG4 (1/100, sc-33666, Santa Cruz, Texas, USA), COL4A1 (1/2000, ab236640, Abcam, UK) and PDGFRB (1/1000, ab32570, Abcam, UK) overnight at 4 °C. The sections were then incubated with a FITC-conjugated rabbit anti-human IgG(H+L) (1/500, A0562, Beyotime Biotechnology, Shanghai, China) or Cy3-conjugated goat anti-rat IgG(H+L) (1/500, A0521, Beyotime Biotechnology, Shanghai, China)

antibody for 1 h at room temperature and mounted using Fluoromount (P0133, Beyotime Biotechnology, Shanghai, China). The slides were examined using a Zeiss LSM710 confocal microscope.

**Tissue histological staining**. Whole eyes and renal cortices of mice were isolated and fixed in 4% paraformaldehyde for 24 h, embedded in paraffin blocks, sectioned and subjected to immunohistochemistry. Kidney sections were processed using standard periodic acid-schiff (PAS) staining, while retina sections were processed using standard haematoxylin and eosin (H&E) staining. Retinal hypotonic isolated vasculature was prepared as previously described. Briefly, the eyes were enucleated, and the retinas were separated under a microscope. Then, the retinas were placed in 1 ml of deionized water in a 24-well plate, shaken at 200 rpm with a 1.5 mm vibration orbit for 1–1.5 h at room temperature and washed a minimum of 3 times with deionized water for 5 min with shaking at 150–300 rpm to remove neuronal cell debris; washing continued until the retinal vasculature appeared to be clear under a microscope. PAS staining revealed the retinal vasculature.

**Retinal fluorescence perfusion**. Sterile fluorescein isothiocyanate FITC-BSA (50 μg/μL) (FD2000S, Sigma-Aldrich, MO, USA) in PBS was injected into the left ventricle of mice at a dose of 100 μg/g body weight. After 2 min, the mice were killed, and the eyes were enucleated and fixed in 4% paraformaldehyde for 30 min. Then, retinas were dissociated under a stereomicroscope and spread on slides. After adding a few drops of anti-fluorescence quenching agent and covering with cover glass, the sections were examined with a Leica microscope.

**ELISA**. Chemokines in the cell culture supernatant were assayed using enzyme-linked immunosorbent assay (ELISA) kits (EK0722, EK0728, EK0359 and EK0413 from BOSTER, Wuhan, China and EK1264 and EK1265 from MULTI SCIENCES, Hangzhou, China) following the manufacturer's instructions.

**Prediction of CKD Progression**. Participants were grouped into 4 groups according to chemokine scores as previously mentioned. Serum creatinine was measured at the time of performing RNA-seq and then annually until 2022, and eGFR was calculated with the CKD Epidemiology Collaboration (CKD-EPI) equation. We discarded the data with eGFR ≤15 ml/min/1.73 m². The association of follow-up time with eGFR decline was assessed using a linear mixed-effects model with both random intercept and slope terms, and the final analyses were adjusted for age and sex.

**Statistics and reproducibility**. The scRNA-seq datasets used in this study included human kidney and retina data from 24 CryoStor preserved human kidney samples and 3 human donor eyes, as mentioned above. For the scRNA-seq analysis of mice, we combined kidney samples from 3 mice and retina samples from 8 mice to obtain cell suspensions. From this, we obtained a total of 21,238 glomerular cells and 9264 retina cells. We also included 29 biopsy-confirmed type 2 DN patients and 20 healthy controls in the study. RNA-seq assays were performed with triplicates for each biological sample. ELISA assays were performed with 4 replicates. qPCR assays were performed using both triplicate biological and technical replicates, and the 2-ΔΔCT method was used for analysis. The chemokines and chemokine scores were compared using an unpaired two-tailed Student's *t* test, while pairwise comparisons among multiple groups were made using one-way ANOVA with Tukey's test. A *p*-value less than 0.05 was considered statistically significant.

**Reporting summary**. Further information on research design is available in the Nature Portfolio Reporting Summary linked to this article.

## Data availability

The raw and processed scRNA-seq and RNA-seq data used in this study have been deposited in the Gene Expression Omnibus (GEO) database with accession number: GSE204880. The human kidney and retina scRNA-seq data was downloaded from GEO database with accession number: GSE140989 and GSE142449. The genome-wide gene expression profiling in glomeruli between DN patients and control donors was downloaded from GEO database with accession number: GSE96804. Source data underlying figures are provided in Supplementary Data.

## Code availability

The R scripts used for analysis and visualization are available upon reasonable request to the corresponding author.

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

## Acknowledgements

This work is supported by the National Natural Science Foundation of China (32141004), the National Natural Science Foundation of China (82070755), and the Open Project of Jiangsu Biobank of Clinical Resources (JSRB2021-01).

## Author contributions

Y.X. developed the methodology, carried out data analysis and wrote the manuscript. Z.X. and W.E. supported the bioinformatics analysis. Y.L., S.H. and W.Q. assisted with performing experiments. J.Y., Z.C. and Z.L. conceived of and supervised the project, and reviewed the manuscript; all authors read and approved the manuscript.

## Competing interests

The authors declare no competing interests.

**Additional information**

