## [Peer Review File · Communications Biology]

Reviewers' comments:

Reviewer #1 (Remarks to the Author):

This manuscript describes the attempt to use single cell RNA approach to assess the link between Diabetic Nephropathy (DN) and Diabetic Retinopathy (DR) by specifically focusing on gene expression correlation between renal mesangial cells (MCs) and retinal pericytes (RPCs). This study used fairly conventional method of scRNA seq on human and mouse renal and retinal tissues to map the makeup of the tissues in presence and absence of diabetes. The main finding is that MCs and RPCs present some similarities in their expression profiles under normal as well as under diabetes conditions. The authors also claim that the effect of diabetes is rather similar and that these findings suggest that using these expression profiles (not accessible in patients – at least not the retinal ones) could help with prognosis of links between DN and DR development. While the finding of similarities between MCs and RPCs is relatively novel and partially supported by the data provided, it is already widely known that a patient with DR or DN is at increased risk of having other complications and thus only provide limited progress to the field. There are also multiple other problems listed below.

Major comments

One of the major issues with this study is the complete lack of information relative to the donors from which tissues were used for the scRNA seq analysis. Absolutely no information is provided relative to the way the tissues were recovered, the timing of recovery, the specifics about the tissues as well as any information relative to the donors history and specifics (demographic, diabetes specifics - duration, A1c,... - , DR grading, DN status, etc...). This issue is also present for the patients used in the last section of the study when the authors are attempting to compare patients with DN, with or without DR, but no specifics are given relative to these patients either.

There are many examples of overstatements of the findings that are not supported by the data and require more nuance in the writing. This is particularly true relative to the main point that the authors are making relative to the similarities between MCs and RPCs. While it is appropriate to point to some striking and important similarities, there are multiple instances of overstatement. For example, relative to Fig 1c, it is untrue to state that "TAGLN, RGS5 and TPM2 were expressed exclusively in MCs and RPCs but not in other cell types" when this figure clearly shows that they are also expressed in FIB (TAGLN) and POD (TPM2). Another such example is relative to "we found that many of the marker genes shared by MCs and RPCs are related to extracellular matrix (ECM) composition, such as Fn1, Cd34 and Itga8." when Fn1 is not expressed in MCs and Itga8 is hardly expressed in RPCs based on the data provided in the corresponding figure.

Some of the validation work is also questionable. The IF in Sup Fig 2 for the expression of RPC marker PDGFRB in cultured MCs and the MC marker ACTA2 in cultured RPCs are very unconvincing and missing some key controls.

Throughout the manuscript and in each of the first 3 figures, the MC and RPC clusters should be annotated separately as it is clear from data in Figure 1 and especially in Figure 2 (2c), that while there are some interesting similarities between MCs and RPCs, there are also some clear differences as shown by only 4 of the 10 genes listed in 2c being really consistently expressed in both cell types. Overall, what figure 2 really demonstrates is that among a subgroup of cells (mostly vascular and some immune cells), they are closer together than from any of the other cells analyzed, and thus need to be more nuanced in its description.

The GESA analysis is interesting but it looks like the changes presented are only based on top pathways in MCs, and don't necessarily match those in RPCs... what does it look the other way around? In other word, the authors should also show the top pathways for RPCs and what they look like in MCs.

It is really unclear to this reviewer what the point is to focus on the impact of AGEs on isolated MCs and RPCs from db/db mice when the authors can focus on these cells in the actual diabetic milieu. This

is further emphasized by the fact that the results of the AGE treatment (fig 4) do not seem to reflect the findings of diabetic vs control as it relates for example to CxCLs (Fig 3).

Other comments

The authors are using some differences between renal and retinal endothelial cells (ECs) to further support the value and importance of the similarities found between MCs and RPCs. While this is interesting and should be noted, this is far less "dramatic" than what the authors make it sound and the associated statements should be dampened.

Make sure to refer to the proper name and nomenclature for all cells (i.e. mullar, most likely instead of Müller cells).

Need to better describe and annotate the figures (supplementary figure 1 for example – lack information about the numbering and abbreviations)

There are also multiple examples of improper English, typos and need for rewording for language accuracy: i.e. "ECs from different tissues were heterogeneous" l 96-97; "suggesting potential consistent molecular transitions" l133; ...

Reviewer #2 (Remarks to the Author):

The authors aim to study kidney and retinal cells together to explain the association between diabetic nephropathy and retinopathy. They show that two mesenchymal cells retinal mesangial cells and retinal pericytes group together, they respond to diabetes similarly with chemokine expression.

Methods can be improved: Show retinal or kidney source of cells in the cluster with MC or RPC in both figure 1 and 2 to exclude this is driven by one cell type. Show before and after integration to exclude changes due to batch effects.

No data on method to integrate the datasets is given. Also, as human data is re-used from ref 18 and 19, please discuss isolation methods and show not-integrated data. What is the source of human data in figure 1: healthy or diabetic or both?

The step from chemokine GSEA from in vitro work towards plasma levels for prediction is quite large. There is no proof that plasma chemokine is actually derived from/expressed on protein level by MC/RPCs in kidney/retina. This could be improved by showing the chemokine score in data of figure 3 in vivo mouse- split by disease state, quantifications of co-staining tissues of healthy and diabetic humans and mice with RGS4/csp4 and chemokines.

minor

Fig 4F y-axis; proportion

Fig 5 legends : text following last panel: urine and eGR are not shown here, please show or remove from legend

Reviewer #3 (Remarks to the Author):

The manuscript by Xu et al, entitled "Single-cell transcriptomes reveal the cellular and molecular linkage between 2 diabetic kidney and retinal lesions", described the single-cell RNA sequencing profiles of kidney and retina and found renal mesangial cells and retinal pericytes shared common molecular features and underwent similar molecular transitions under diabetes. The authors further uncovered that chemokine up-regulation shared by the two cell types were critical for the co-occurrence of nephropathy (DN) and diabetic retinopathy (DR), implying that the chemokine score could be used for the prognosis of DN complicated with DR.

The most convincing evidence is from the data in Figure 1a, showing that mesangial cells (MC) and retinal pericytes (RPC) share the same cluster.

Major Critiques:

1. The authors indicated that TAGLN, RGS5 and TPM2 were expressed exclusively in MCs and RPCs but not in other cell types, but these genes expression was actually present in other cell types (Figure 1d).
2. The clusters of the single-cell RNA-seq are not well differentially separated.
3. No biological evidence that their candidates indeed are important for progression of DN And DR.

Minor Critiques:

1. In Figure 4, proportion need to be changed to proportion

Response Letter

Reviewer 1:

This manuscript describes the attempt to use single cell RNA approach to assess the link between Diabetic Nephropathy (DN) and Diabetic Retinopathy (DR) by specifically focusing on gene expression correlation between renal mesangial cells (MCs) and retinal pericytes (RPCs). This study used fairly conventional method of scRNA seq on human and mouse renal and retinal tissues to map the makeup of the tissues in presence and absence of diabetes. The main finding is that MCs and RPCs present some similarities in their expression profiles under normal as well as under diabetes conditions. The authors also claim that the effect of diabetes is rather similar and that these findings suggest that using these expression profiles (not accessible in patients – at least not the retinal ones) could help with prognosis of links between DN and DR development. While the finding of similarities between MCs and RPCs is relatively novel and partially supported by the data provided, it is already widely known that a patient with DR or DN is at increased risk of having other complications and thus only provide limited progress to the field. There are also multiple other problems listed below.

Major comments

Comment 1: One of the major issues with this study is the complete lack of information relative to the donors from which tissues were used for the scRNA seq analysis. Absolutely no information is provided relative to the way the tissues were recovered, the timing of recovery, the specifics about the tissues as well as any information relative to the donors history and specifics (demographic, diabetes specifics - duration, A1c,... , DR grading, DN status, etc...). This issue is also present for the patients used in the last section of the study when the authors are attempting to compare patients with DN, with or without DR, but no specifics are given relative to these patients either.

Response 1: Thank you for your valuable comments. The human donor scRNA-seq datasets analyzed here were obtained from the publicly available GEO datasets. We specifically selected samples from these datasets that were derived from healthy individuals. We have added the information of these healthy controls in the revised version lines 345-362: The output files of the Cell Ranger (10x Genomics) pipeline of healthy human kidney (GSE140989) and expression matrix files of healthy human retina (GSE142449) were obtained from the NIH GEO dataset. Kidney samples were composed of 24 CryoStor preserved human kidney samples: 16 tumor-nephrectomy, 5 human allograft kidney transplant surveillance, and 3 preperfusion transplant biopsy samples. Kidney tissues were enzymatically digested at 37 °C after resuscitation. Human donor eyes ($n = 3$) were obtained through the Iowa Lions Eye Bank and were received in the laboratory within 5.5 hours after death and were immediately subjected to enzyme dissociation at 37 °C immediately and then cryopreserved. Both dissociated cells were processed with 10x Genomics and the Illumina HiSeq 4000 platform. The expression files were

transformed to Seurat objects through the “CreateSeuratObject” function. Both datasets were normalized and scaled and PCA was performed using highly variable genes that were selected using the Seurat function “FindVariableFeatures”. Cells were annotated with marker genes provided by original articles. To perform batch correction and data integration, an anchor was created via the Seurat function “FindIntegrationAnchors” and then integrated through the Seurat function “IntegrateData”. The data were then scaled, analysed for PCA, and visualized using UMAP.

The information of patients used in the last section of results was detailed in lines 364-372: Human glomeruli were obtained from the National Clinical Research Center of Kidney Diseases, Nanjing, China. The glomeruli were microdissected and subjected to RNA extraction, followed by cDNA synthesis and genome-wide gene expression analysis using the Affymetrix® microarray platform (HTA 2.0), as we previously described³⁰. Chemokine score analysis included 29 biopsy-confirmed type 2 DN patients with fundus examination results and 20 healthy controls. Among all the DN patients, 14 individuals were identified as having DN complicated with DR, whereas the other 15 patients did not exhibit fundus changes. The clinical features of these patients are listed in Supplementary Table. 2.

Supplementary Table. 2 Clinical characteristics of DN patients complicated without and with DR.

	DN with DR patients (n = 14)	DN without DR patients (n = 15)	P value
Age (years)	46.373±8.299	47±8.058	0.8383
BMI (kg/m ²)	26.182±2.182	24.31±2.44	0.0381
HbA1c (%)	7.05±0.924	7.257±1.621	0.6732
eGFR(ml/min/1.73m ²)	75.447±30.304	58.015±26.867	0.1138
Proteinuria (g/24h)	2.152±3.422	4.385±3.456	0.0920
SBP (mmHg)	138.2±15.025	155.929±24.687	0.0261
DBP (mmHg)	81.6±10.986	89.429±11.673	0.0737

Comment 2: There are many examples of overstatements of the findings that are not supported by the data and require more nuance in the writing. This is particularly true relative to the main point that the authors are making relative to the similarities between MCs and RPCs. While it is appropriate to point to some striking and important similarities, there are multiple instances of overstatement. For example, relative to Fig 1c, it is untrue to state that “TAGLN, RGS5 and TPM2 were expressed exclusively in MCs and RPCs but not in other cell types” when this figure clearly shows that they are also expressed in FIB (TAGLN) and POD (TPM2). Another such example is relative to “we found that many of the marker genes shared by MCs and RPCs are related to extracellular matrix (ECM) composition, such as Fn1, Cd34 and Itga8.” when Fn1 is not expressed in MCs and Itga8 is hardly expressed in RPCs based on the data provided in the corresponding figure.

Response 2: Thank you for pointing this out. We have checked the manuscript and corrected the overstatement, as shown in lines 93-94 that expression of *RGS5*, *ACTA2* and *MYH11* were shared mainly in MCs and RPCs but rarely in other cell types (Fig. 1d),

lines 122-124, “Furthermore, we found that many of the marker genes shared by MCs and RPCs are related to extracellular matrix (ECM) composition, such as *Cd34*, *Cspg4* and *Mcam*”.

Comment 3: Some of the validation work is also questionable. The IF in Sup Fig 2 for the expression of RPC marker PDGFRB in cultured MCs and the MC marker ACTA2 in cultured RPCs are very unconvincing and missing some key controls.

Response 3: Thank you for your valuable questions. We have repeated the experiment and used the IgG isotype as a control, as shown in the new Supplementary Fig. 2.

Comment 4: Throughout the manuscript and in each of the first 3 figures, the MC and RPC clusters should be annotated separately as it is clear from data in Figure 1 and especially in Figure 2 (2c), that while there are some interesting similarities between MCs and RPCs, there are also some clear differences as shown by only 4 of the 10 genes listed in 2c being really consistently expressed in both cell types. Overall, what figure 2 really demonstrates is that among a subgroup of cells (mostly vascular and some immune cells), they are closer together than from any of the other cells analyzed, and thus need to be more nuanced in its description.

Response 4: We appreciate your kind suggestions. The clusters of MC and RPC are annotated separately in Fig. 2a and 3a.

Fig. 2a UMAP plot of cells from glomeruli and retinal vessels. The cell types and sample types of origin are annotated on the right.

Fig. 3a UMAP plot of cells of glomeruli and retinal vessels from 21-week-old db/db mice. The cell types and sample types of origin are annotated on the right.

The illustration of Fig. 2 has been changed in lines 114-116: The results showed that MCs and RPCs were grouped together more than any of the other cell types analysed, indicating that MCs and RPCs are more similar to each other than to other vascular cell types.

Comment 5: The GESA analysis is interesting but it looks like the changes presented are only based on top pathways in MCs, and don't necessarily match those in RPCs... what does it look the other way around? In other word, the authors should also show the top pathways for RPCs and what they look like in MCs.

Response 5: Thank you for your suggestion. We further showed the top pathways of MCs and RPCs in Fig. 3c, which was illustrated in lines 135-140: When we performed Gene Set Enrichment Analysis (GESA) to study the changes in these cells under diabetic conditions, we found that both MCs and RPCs upregulated the expression of genes related to the signaling pathways of ECM-receptor interaction and PI3K-Akt signaling pathway, while some energy metabolism pathways such as TCA cycle were upregulated only in RPCs (Fig. 3c).

Fig. 3c GESA results show the upregulated pathways in MC and RPC under diabetic conditions.

Comment 6: It is really unclear to this reviewer what the point is to focus on the impact of AGEs on isolated MCs and RPCs from db/db mice when the authors can focus on these cells in the actual diabetic milieu. This is further emphasized by the fact that the results of the AGE treatment (fig 4) do not seem to reflect the findings of diabetic vs control as it relates for example to CxCLs (Fig 3).

Response 6: Thank you for your comments. We conducted the cell experiments *in vitro* for the following reasons: The internal environment is relatively complex, and for specific cell types, they can be interfered with by many other factors, such as compensatory mechanisms and interactions with other cells. Cultured MCs and RPCs under AGEs treatment may reflect simple and direct cell lesions in diabetes. *In vitro*, we observed the upregulation of multiple CXCLs, while in Fig. 3c, although the chemokine signaling pathway did not rank high, we also observed an upregulation of this pathway, especially in MCs. The alterations in the chemokine signaling pathway observed in RPCs were not significant when studied *in vivo*. It cannot be excluded that RPCs are affected *in vivo* by other factors mentioned above. We revised the discussion and provide a more objective interpretation of the *in vitro* experimental results. In addition, we administered AGEs to cultured human MCs and RPCs, which could compensate for our deficiencies in using mouse models.

Other comments

Comment 1: The authors are using some differences between renal and retinal endothelial cells (ECs) to further support the value and importance of the similarities found between MCs and RPCs. While this is interesting and should be noted, this is far less “dramatic” than what the authors make it sound and the associated statements should be dampened.

Response 1: We sincerely appreciate the valuable comments. To make the expression more appropriate, we have revised lines 88-91 as follows: Although ECs were present in both the kidney and retina, we found that EC-GCs and EC-Rs were separated and clustered into different subsets, indicating tissue-specific heterogeneity of ECs. In contrast, we found that MCs clustered together with RPCs.

Comment 2: Make sure to refer to the proper name and nomenclature for all cells (i.e. mullar, most likely instead of Müller cells).

Response 2: Thank you for your careful examination. We have checked the cell type names and changed the wrong name mullar to müller cells.

Comment 3: *Need to better describe and annotate the figures (supplementary figure 1 for example – lack information about the numbering and abbreviations)*

Response 3: We have checked all the descriptions and annotations of the figures, including Supplementary Fig. 1.

Comment 4: There are also multiple examples of improper English, typos and need for rewording for language accuracy: i.e. “ECs from different tissues were heterogeneous” heterogenous; l 96-97; “suggesting potential consistent molecular transitions” l133; ...

Response 4: Thank you for your careful examination. We have carefully checked and improved the English writing in the revised manuscript.

Reviewer 2:

The authors aim to study kidney and retinal cells together to explain the association between diabetic nephropathy and retinopathy. They show that two mesenchymal cells, retinal mesangial cells and retinal pericytes, group together; they respond to diabetes similarly with chemokine expression.

Comment 1: Methods can be improved: Show retinal or kidney source of cells in the cluster with MC or RPC in both figure 1 and 2 to exclude this is driven by one cell type. Show before and after integration to exclude changes due to batch effects.

Response 1: Thank you for your valuable advice. We have added the retinal or kidney source of cells and the cluster distribution before integration, as shown in Fig. 1a, Fig. 2a and Fig. 3a.

Fig. 1a UMAP plot of 22,262 renal cells and 9,711 retinal cells colour-coded according to the number of clusters. The clusters annotated as MCs, RPCs, ECs, POD and FIB are highlighted and the sample types of origin are shown.

Fig. 2a UMAP plot of cells from glomeruli and retinal vessels. The cell types and sample types of origin are annotated on the right.

Fig. 3a UMAP plot of cells of glomeruli and retinal vessels from 21-week-old db/db mice. The cell types and sample types of origin are annotated on the right.

Comment 2: No data on method to integrate the datasets is given. Also, as human data is re-used from ref 18 and 19, please discuss isolation methods and show not-integrated data. What is the source of human data in figure 1: healthy or diabetic or both?

Response 2: Thank you for your kind reminder. We have added the data source, isolation and integration methods in Methods, lines 345-362: The output files of the CellRanger (10x Genomics) pipeline of **healthy** human kidney (GSE140989) and expression matrix files of **healthy** human retina (GSE142449) were obtained from the NIH GEO dataset. Kidney samples were composed of 24 CryoStor preserved human kidney samples: 16 tumor-nephrectomy, 5 human allograft kidney transplant surveillance, and 3 preperfusion transplant biopsy samples. Kidney tissues were enzymatically digested at 37 °C after resuscitation. Human donor eyes ($n = 3$) were obtained through the Iowa Lions Eye Bank and were received in the laboratory within 5.5 hours after death and were immediately subjected to enzyme dissociation at 37 °C immediately and then cryopreserved. Both dissociated cells were processed with 10x Genomics and the Illumina HiSeq 4000 platform. The expression files were transformed to Seurat objects through the "CreateSeuratObject" function. Both datasets were normalization and scaled and PCA was performed using highly variable genes that were selected using the Seurat function "FindVariableFeatures". Cells were annotated with marker genes provided by original articles. To perform batch correction and data integration, an anchor was created via the Seurat function "FindIntegrationAnchors" and then integrated through the Seurat function "IntegrateData". The data were then scaled, analysed for PCA, and visualized using UMAP.

Comment 3: The step from chemokine GSEA from in vitro work towards plasma levels for prediction is quite large. There is no proof that plasma chemokine is actually derived from/expressed on protein level by MC/RPCs in kidney/retina. This could be improved by showing the chemokine score in data of figure 3 in vivo mouse- split by disease state, quantifications of co-staining tissues of healthy and diabetic humans and mice with RGS4/csp4 and chemokines.

Response 3: Thank you for the valuable suggestion. Our research focused on MCs and RPCs, and we found the common upregulation of chemokines, which could be a potential predictor of DN complicated with DR. We have realized that it is too broad to say that chemokines are expressed only in MCs and RPCs and we have revised relative statements.

To provide biological evidence, we calculated chemokine scores of MCs and RPCs in vivo (Fig. 4f) and co-staining kidneys of healthy and diabetic humans and mice with RGS4 and chemokines, including CXCL1 and CXCL3 (Fig. 5).

Fig. 4f Elevated chemokine scores of MC and RPC in wt and *db/db* mice.

Fig.5 Verification of the increased expression of chemokines in DN by co-staining tissues of healthy and diabetic humans and mice with the MC and RPC markers CSPG4 and chemokines. a Co-staining of CSPG4 and CXCL1 in the kidney of wt and *db/db* mouse and in the kidney of normal human and diabetic patient. b Co-staining of CSPG4 and CXCL3 in the kidney of wt and *db/db* mouse and in the kidney of normal human and diabetic patient.

minor

Comment: Fig 4F y-axis; propOrtion

Fig 5 legends: text following last panel: urine and eGR are not shown here, please show or remove from legend

Response: We apologize for our careless mistakes. Thank you for your reminder, and we have corrected the mistakes.

Reviewer 3:

The manuscript by Xu et al, entitled “Single-cell transcriptomes reveal the cellular and molecular linkage between 2 diabetic kidney and retinal lesions”, described the single-cell RNA sequencing profiles of kidney and retina and found renal mesangial cells and retinal pericytes shared common molecular features and underwent similar molecular transitions under diabetes. The authors further uncovered that chemokine up-regulation shared by the two cell types were critical for the co-occurrence of nephropathy (DN) and diabetic retinopathy (DR), implying that the chemokine score could be used for the prognosis of DN complicated with DR.

Comment 1: The authors indicated that TAGLN, RGS5 and TPM2 were expressed exclusively in MCs and RPCs but not in other cell types, but these genes expression was actually present in other cell types (Fig. 1d).

Response 1: We sincerely thank you for careful reading. The cells in red circle next to endothelial cell clusters belong to cluster 17, and these cells express marker genes for MCs and RPCs (Fig. 1d). From their gene expression profile and the inappropriate localization, we hypothesized that they might be cell doublets. We performed DoubletFinder (PMID: 30954475) and demonstrated that there are many doublets in this area, which were removed in our revised version.

Then we obtained the new version of Fig. 1.

Fig. 1a UMAP plot of renal cells and retinal cells colour-coded according to the number of clusters. The clusters annotated as MC, RPC, EC, POD and FIB are highlighted and the sample types of origin are shown.

Comment 2: The clusters of the single-cell RNA-seq are not well differentially separated.

Response 2: We performed single-cell RNA-seq data processing according to methods used in the published studies (PMID: 35361264, PMID: 32398875). The filtered expression matrix was normalized with the function “NormalizeData”, followed by the identification of 2000 genes of high cell-to-cell variation by using the function “FindVariableFeatures”. We then performed principal component analysis (PCA) with the top 2000 variable features by using the function “RunPCA”. Cells were then clustered using the functions “FindNeighbors” and “FindClusters” with the first 50 principal components (PCs). Finally, UMAP was performed on the top 50 PCs by using the function “RunUMAP” for nonlinear dimensional reduction and data visualization. To make clusters separated better, we made some effort, such as the use of umap instead of tsne. We supposed that vascular and immune cells captured in our study are relatively close in physical location or function and consequently difficult to separate well.

Comment 3: No biological evidence that their candidates indeed are important for progression of DN And DR.

Response 3: Thank you very much for your comments. This study was a retrospective analysis and the case data lacked the grading of DR. Regarding the progression of DN, we collected the serum creatinine annually and calculated estimated glomerular filtration

rate (eGFR) with the CKD Epidemiology Collaboration (CKD-EPI) equation. By performing a linear mixed-effects model, we found that eGFR decreased more rapidly with increasing chemokine score and that patients with high chemokine score entered replacement therapy more rapidly and the corresponding eGFR is missing (Supplementary Fig. 5).

Supplementary Fig. 5 Calibration plots of observed and linear fitting for DN patients grouped by chemokine scores. DN patients with high chemokine scores show rapid declines in eGFR.

Minor Critiques:

Comment 1: In Figure 4, proportion need to be changed to proportion

Response 1: Thank you for your kind reminder, we have corrected that spelling error.

Reviewers' comments:

Reviewer #1 (Remarks to the Author):

The authors have addressed a lot of the original comments however they are still far over-reaching at times with their interpretation and associated statements.

It is particularly still the case for the results shown in figure 5 and the co-staining of CXCL1/CXCL3 with CSPG4 in kidney from db/db mice and from human donors. The authors state that the images denote partial colocalization when they almost show a complete opposite staining pattern, especially for CXCL3.

Overall, there remains an issue with the amount of information provided relative to the patients analyzed. It is important to know what the level/type of DR (mild/moderate/severe NPDR/PDR/DME...) was in the patients studied to make the observation of potential correlation and "link" between DN and DR (well known already) and the "cytokine score" (less well define) really valuable.

There are also still some examples of improper English, duplicated words, typos and need for rewording for language accuracy (analysed instead of analyzed; phycological instead of physiological, immediately repeated twice, etc...).

Overall, the main finding remains some similarities between MCs and RPCs, much less striking than originally described (even if supported by some of the data provided), but it is already wildly known that a patient with DR or DN is at increased risk of having other complications and thus only provide limited progress to the field relative to the potential of the "chemokine score" as a link between DR and DN.

Reviewer #2 (Remarks to the Author):

Authors clarified integration methods. The authors state they are showing data in 1a and 2a before integration, while I aske din point 1 to show both before and after. Im not sure if this is indeed plots before integration, but In any case it doesnt seem like the cells ar clustered by celltype, rather by source. in that case integration is not very good, and authors may try with different methods., like Harmony. This is important, considering the two completely separate data sources and generation, adn teh conclusosn taht cells are very similar. Otherwise maybe the authors were confused by my remark in point 2, asking for data before integration (to be able to compare to data in the paper, which I assumed to be "after integration").

other points are well addressed

Reviewer #3 (Remarks to the Author):

None

Response Letter

Reviewer 1:

Comment 1: The authors have addressed a lot of the original comments however they are still far over-reaching at times with their interpretation and associated statements. It is particularly still the case for the results shown in figure 5 and the co-staining of CXCL1/CXCL3 with CSPG4 in kidney from db/db mice and from human donors. The authors state that the images denote partial colocalization when they almost show a complete opposite staining pattern, especially for CXCL3.

Response 1: Thank you for pointing this out. In our previous revision, we attempted to provide biological evidence that chemokines are expressed at protein level in mesangial cells in kidney. In the figures showing the co-staining of mesangial cell marker CSPG4 with chemokines CXCL1/CXCL3, we used equilateral triangle that may generate misleading indications, and now we use new indicating arrows to show co-staining cells as figure below.

Verification of the increased expression of chemokines in DN by co-staining tissues of healthy and diabetic humans and mice with the MC markers CSPG4 and chemokines CXCL1/CXCL3.

We found increased expression of CXCL1 and CXCL3 in glomeruli including mesangial cells in *db/db* mice and DN patients compared with controls. Due to the weak expression of CXCL3 and its expression in renal tubule, its association with mesangial cells might be

debatable and less compelling, as you pointed out in your comment. Consequently, we deleted the results of co-staining of CXCL3 and CSPG4 in this revised version.

Comment 2: Overall, there remains an issue with the amount of information provided relative to the patients analyzed. It is important to know what the level/type of DR (mild/moderate/severeNPDR/PDR/DME...) was in the patients studied to make the observation of potential correlation and “link” between DN and DR (well known already) and the “cytokine score” (less well define) really valuable.

Response 2: Thank you for your valuable comments. Firstly, we cannot agree more with your viewpoint that the grading of DR is important and the level/type of DR was significant to make the observation of potential correlation between DN and DR. We have described in method that patients with fundus examination confirmed to have fundus lesions were included as DR patients. Those patients were hospitalized in the nephrology department, and their fundus examination results are based on the relatively simple description recorded in the inpatient medical record written by ophthalmologist other than formal inspection report. For example, the fundus examination result of patient 1: scattered bleeding in the fundus of the eyes; patient 2: clear boundary on fundus examination, disappearance of macular fovea reflection, retinal hard exudation, visible vascular tortuosity in the left eye, suspicious neovascularization, retinal arteriovenous ratio of 1:2, and degree of arteriosclerosis of grade 2. Therefore, it was hard to classify the accurate level of DR. Considering that the cohort was collected several years ago, and with disease progression, it is difficult to collect fundus data from these patients for grading now. This is also what we regret.

Comment 3: There are also still some examples of improper English, duplicated words, typos and need for rewording for language accuracy (analysed instead of analyzed; phycological instead of physiological, immediately repeated twice, etc...).

Response 3: Thank you for your careful reading. We have revised the improper English: analysed is the spelling of British English and we have corrected to analyzed in line 114,115, etc. And we corrected phycological to physiological in line 70 and deleted immediately in line 353.

Comment 4: Overall, the main finding remains some similarities between MCs and RPCs, much less striking than originally described (even if supported by some of the data provided), but it is already wildly known that a patient with DR or DN is at increased risk of having other complications and thus only provide limited progress to the field relative to the potential of the “chemokine score” as a link between DR and DN.

Response 4: Thank you for your comments. First of all, that is well known that a patient with DR or DN is at increased risk of having other complications, which is the basis of our research on this topic. We also conducted information retrieval, trying to elucidate the mechanism of this phenomenon from pathogenesis, pathological changes, and so on. We found that the emergence of single cell transcriptome technology has opened up new opportunities, and has unprecedented advantages in interpreting cell type specific changes. Therefore, we attempted to use single cell transcriptome to explain the underlying

mechanism of the relationship between DN and DR. The pathological changes of DN and DR are manifested at the organ level, but we have to admit that organs are composed of several or even dozens of cell types, which have different or even completely opposite responses to diseases. Prior to our study, no study has explored the mechanism of the correlation between DN and DR at single cell level. Admittedly, the chemokine score may not be immediately applicable to clinical work, but our study provides new ideas and insights for future research, such as whether the increase of chemokines will lead to the infiltration of immune cells in the kidney and retina and thus has a critical impact, and whether the role of mesangial cells and retinal pericytes in DN and DR is underestimated.

Reviewer 2:

Comment 1: Authors clarified integration methods. The authors state they are showing data in 1a and 2a before integration, while I asked in point 1 to show both before and after. I'm not sure if this is indeed plots before integration, but in any case, it doesn't seem like the cells are clustered by cell type, rather by source. in that case integration is not very good, and authors may try with different methods., like Harmony. This is important, considering the two completely separate data sources and generation, and the conclusion that cells are very similar. Otherwise maybe the authors were confused by my remark in point 2, asking for data before integration (to be able to compare to data in the paper, which I assumed to be "after integration").

Response 1: Thank you for your comments and we apologize for not fully comprehending your comments in the last revision. This is the UMAP plot from adult kidney in PMID: 32107344 and retina in PMID: 32069977 (Among them, the cells of healthy controls were extracted and used in our study).

UMAP plot from adult kidney in PMID: 32107344 and retina in PMID: 32069977.

After we downloaded the expression matrix of healthy human kidney and retina from GSE datasets, we performed UMAP dimensionality reduction process and got the following plots:

UMAP plot of healthy adult kidney derived from PMID: 32107344 and retina from PMID: 32069977.

In our study, we used Seurat 3 to integrate the cells of kidney and retina, following the instruction of Nat Biotechnol. 2018 (PMID: 29608179), which was a widely used integration method in scRNA-seq. Building on your suggestion, we attempted another integration methods Harmony, and got the following plots:

UMAP plots of raw data, Harmony, Seurat 3 outputs, the cells in left panel are colored by source, and in the right panel by cell type.

In Harmony, MCs and RPCs are grouped together, as found in Seurat 3. Therefore, we added the result of harmony in supplementary figure 2. We also found that EC subclusters show better group boundaries and less dispersion in Seurat 3. Leaving aside integration methods, we have to admit that kidney and retina are two organs with linkage but still many cell types absence in the other, due to this reason, We cannot completely mix them perfectly, as if they are the same organ.

Reviewers' comments:

Reviewer #1 (Remarks to the Author):

The authors have now addressed all but one comment in the manuscript itself. Since the authors explained that the information on the exact DR status are not known due to how they were collected, they should clarify that point in the method section. They should also briefly explain the limitations associated and discuss this point in the results and discussion section so the readers can take that into consideration when looking at the data.

Reviewer #2 (Remarks to the Author):

the authors provide new integration of two datasources based on harmony. This looks good for MCs and RPCs, in contrast to seurat integration. so all downstream analysis should be based on Harmony and redone, as otherwise the DEGs are based on differences between sources. Simply putting Harmony in the supplements is not sufficient, as seurat approach is incorrect and thus method flawed. Unless the authors can show there is considerable overlap between data stemming from seurat and Harmony objects

Response Letter

Reviewer #1:

The authors have now addressed all but one comment in the manuscript itself. Since the authors explained that the information on the exact DR status are not known due to how they were collected, they should clarify that point in the method section. They should also briefly explain the limitations associated and discuss this point in the results and discussion section so the readers can take that into consideration when looking at the data.

Response: Thank you for pointing this out. We have clarified the limitation associated in line 186 in the result section, line 259-263 in the discussion section and line 376-378 in the method section.

Reviewer #2:

the authors provide new integration of two data sources based on harmony. This looks good for MCs and RPCs, in contrast to seurat integration. so all downstream analysis should be based on Harmony and redone, as otherwise the DEGs are based on differences between sources. Simply putting Harmony in the supplements is not sufficient, as seurat approach is incorrect and thus method flawed. Unless the authors can show there is considerable overlap between data stemming from seurat and Harmony objects.

Response: Thank you for your valuable comments. We have conducted new integration using the Harmony algorithm and performed downstream analysis accordingly. We have also revised the relevant content in the manuscript and updated figures 1-3. For the differential gene analysis in this study, we utilized the DEsingle R package, specifically designed for identifying differentially expressed genes between cell groups in single-cell RNA-seq data (PMID: 29688277), instead of the embedded FindMarkers function in Seurat. Therefore, it is important to note that our differential gene analysis results are independent of source differences, and the integration method change to Harmony does not impact the identification of DEGs.

REVIEWERS' COMMENTS:

Reviewer #2 (Remarks to the Author):

authors have addressed all concerns